# Nanoengineering room temperature ferroelectricity into orthorhombic SmMnO$_3$ films

Eun-Mi Choi [1,5,7✉], Tuhin Maity [1,6,7✉], Ahmed Kursumovic[1], Ping Lu [2], Zenxhing Bi[3], Shukai Yu[4], Yoonsang Park[4], Bonan Zhu [1], Rui Wu[1], Venkatraman Gopalan[4], Haiyan Wang [3] & Judith L. MacManus-Driscoll[1]

Orthorhombic $R$MnO$_3$ ($R$ = rare-earth cation) compounds are type-II multiferroics induced by inversion-symmetry-breaking of spin order. They hold promise for magneto-electric devices. However, no spontaneous room-temperature ferroic property has been observed to date in orthorhombic $R$MnO$_3$. Here, using 3D straining in nanocomposite films of (SmMnO$_3$)$_{0.5}$((Bi,Sm)$_2$O$_3$)$_{0.5}$, we demonstrate room temperature ferroelectricity and ferromagnetism with $T_{C,FM}$ ~ 90 K, matching exactly with theoretical predictions for the induced strain levels. Large in-plane compressive and out-of-plane tensile strains (−3.6% and +4.9%, respectively) were induced by the stiff (Bi,Sm)$_2$O$_3$ nanopillars embedded. The room temperature electric polarization is comparable to other spin-driven ferroelectric $R$MnO$_3$ films. Also, while bulk SmMnO$_3$ is antiferromagnetic, ferromagnetism was induced in the composite films. The Mn-O bond angles and lengths determined from density functional theory explain the origin of the ferroelectricity, i.e. modification of the exchange coupling. Our structural tuning method gives a route to designing multiferroics.

[1] Department of Materials Science and Metallurgy, University of Cambridge, Cambridge, UK. [2] Sandia National Laboratories, Albuquerque, NM 87185, USA. [3] School of Materials Engineering, Purdue University, West Lafayette, IN, USA. [4] Department of Physics, The Pennsylvania State University, University Park, State College, PA 16802, USA. [5] Present address: Center for Integrated Nanostructure Physics (CINAP), Institute for Basic Science (IBS), Sungkyunkwan University (SKKU), Suwon 16419, Korea. [6] Present address: School of Physics, Indian Institute of Science Education and Research Thiruvananthapuram, Kerala, India. [8] These authors contributed equally: Eun-Mi Choi, Tuhin Maity. ✉email: emchoi@skku.edu; tuhin@iisertvm.ac.in

Type-II multiferroic materials have exquisitely coupled magnetic and ferroelectric orders and are interesting for future magnetoelectric devices for non-volatile memory[1,2] and sensing applications. In type II orthorhombic rare-earth manganites ($o$-$R$MnO$_3$) ferroelectricity is induced by inversion-symmetry breaking of the magnetic order through the Dzyaloshinskii-Moriya interaction[3]. $o$-$R$MnO$_3$ has a very rich functional phase diagram due to a changing magnetic spin state, from an A-type antiferromagnet (A-AFM) to an E-type anti-ferromagnet (E-AFM) and electrically from paraelectric (PE) to ferroelectric (FE)[3]. However, the ferroic order in $o$-$R$MnO$_3$ originating from cycloidal spiral spin order occurs at a very low temperature, typically below 40 K for bulk and below 100 K for thin films, which makes it impractical for applications. In addition, the electric polarisation ($P$) in these materials is much smaller ($P < 0.1\,\mu$C cm$^{-2}$) compared to Bi-based FE originating from the ordering of lone pairs, even at very low temperature[1,4–9]. For device applications such as energy efficient non-volatile random access memory (RAM) (whether ferroelectric RAM or multistate multiferroics RAM), a high $P$ ($>1\,\mu$C cm$^{-2}$) is strongly desired at room temperature (RT) and above[10].

In terms of the magnetic properties, low-temperature ferro-magnetism with magnetisation ($M$) values near $\sim 1\mu_B$ Mn$^{-1}$ have been reported in thin $o$-$R$MnO$_3$ films[11,12]. Ferromagnetic (FM) ordering in $o$-$R$MnO$_3$ originates due to epitaxial strain which changes the balance between AFM and FM interactions[13,14] and breaks the long range AFM order at the boundary of different domains[5,15]. Spin-driven ferroelectricity with large FE polarisation is expected with large Mn–O–Mn bond angles and small Mn–O bond lengths[16]. Hence, structural distortion of $o$-$R$MnO$_3$ could produce a RT FE-FM multiferroic. According to the Goodenough-Kanemori (GK) rules, a large bond angle will destroy the E-AFM ordering (collinear up-up-down-down: ↑↑↓↓) and simultaneously stabilise a FM phase, but will also lead to a non-collinear spin configuration, giving a low $P$ or a PE A-AFM[16,17]. Therefore, achieving multiferroicity requires a delicate balance of bond angle and bond length tuning. Such tuning has not been demonstrated to date, and hence there are no reports of RT ferroelectricity in $o$-$R$MnO$_3$[13,14]. The highest reported FE transition temperature ($T_{C,FE}$) in $o$-$R$MnO$_3$ films is below $\sim 75$ K and the highest spin-driven FE polarisation is $P \sim 1.5\,\mu$C cm$^{-2}$ under high pressure and high magnetic field[5,18]. At the same time the highest $T_{C,FM}$ of the FM phase is $\sim 105$ K[5]. Recently, however, theoretical and experimental results have demonstrated the pos-sibility of the coexistence of spontaneous FM and enhanced FE polarisation in $R$MnO$_3$ arising from structural distortions[5,17,19]. From theoretical calculations, Iusan et al.[17] reported that under compressive strain ($\sim -4\%$), the AFM phase of $R$MnO$_3$ is not stable and FM ordering emerges with highly enhanced polarisa-tion. However, a $-4\%$ strain is very difficult to achieve in $R$MnO$_3$ thin films using epitaxial strain from the substrate.

Considering the fundamental theoretical works which promise to achieve RT ferroic properties in single phase $o$-$R$MnO$_3$, here we have designed and demonstrated self-assembled vertically aligned nanocomposite (VAN) thin films of SmMnO$_3$ (SMO) + (Bi,Sm)$_2$O$_3$ (BSO), where BSO forms nanocolumns in a SMO matrix. VAN films represent a unique way to create 3D strain[20] and they have several advantages for tuning strongly correlated systems. It is possible to tune both the in-plane and out-of-plane strain independently, thus giving another degree of freedom for bond length and angle tuning[21]. There is no intrinsic thickness limitation to the strain tuning[20]. Very uniform and high strain states can be engineered into the self-assembled VAN films[22].

Of the different $o$-$R$MnO$_3$ phases, SMO is of particular interest since according to theoretical calculations of exchange coupling, a transition from A-AFM to E-AFM in SMO is possible due to

strain or chemical pressure as it sits close to the phase transition in the phase diagram[16]. Hence, a very small perturbation in the bond angle or length is able to significantly modify the magnetic ordering of SMO. The relatively large Mn–O–Mn bond angle in SMO compared to other $o$-$R$MnO$_3$ gives a higher possibility to achieve FM ordering by modifying the FM in-plane nearest and AFM in-plane next-nearest neighbour exchange interactions of Mn moments, $J_1$ and $J_2$, respectively, and in doing so, to induce ferroelectricity. Hence, owing to the type II nature of the multi-ferroicity, if the magnetic properties are modified via strain, the FE properties should also be readily modified.

With the appropriate nanopillars, the VAN system can induce 3D strain into SMO. In this work, we choose BSO as the nanopillar phase owing to the fact that any Bi substitution into SMO should not be strongly detrimental to the magnetic or FE properties of SMO, and also because of the relatively low melting point of Bi$_2$O$_3$ which should mean high crystalline perfection of the pillars. Also, the relatively higher stiffness of BSO compared to SMO[21,23–26] means the BSO should control the strain in the SMO.

In our VAN SMO:BSO system, we find an increase of the FE transition temperature to above RT, which compares to $T_{C,FE}$ $< 40$ K in the bulk. Also, a FM transition temperature, $T_{C,FM}$, of $\sim 90$ K, and saturation moment of $1.02\,\mu_B$ Mn$^{-1}$ at 10 K are obtained without any external pressure or field which compares to the bulk which is AFM, $T_{C,AFM}$ $\sim 60$ K. The RT ferroelectricity in the film is confirmed by using piezoresponse force microscopy (PFM), positive-up and negative-down (PUND) FE pulse tests, and second harmonic generation (SHG) measurements. The net switching polarisation ($2P_R$ where $P_R$ = remnant polarisation) and piezoresponse amplitude ($d_{33}$) are $3.9\,\mu$C cm$^{-2}$ and $6.7$ pm V$^{-1}$, respectively. In addition, long-term retention of polarisation from PFM at RT shows the stable ferroelectricity. SHG polar plots indicate a breaking of centrosymmetry of SMO. The spontaneous RT ferroelectricity with high $T_{C,FM}$ ferromagnetism in SMO is consistent with the presence of a unique strain state in the VAN films. This is proven by growing VAN films of different thick-nesses, and by showing that only the thicker, more highly strained VAN films contain the FE-FM phase. In the thicker VAN films, the nanopillars rather than the substrate control the strain state. Density Functional Theory (DFT) calculations of the Mn–O bond angles and lengths indicate strong exchange coupling, which explains the change in spin state and hence the ferromagnetism and thus RT ferroelectricity. Overall, our work shows a route to achieving high temperature multiferroics using a simple 3D strain approach.

## Results

**Growth of SMO:BSO VAN**. Three different thin films were deposited by pulsed laser deposition (PLD) onto (001) SrTiO$_3$ (STO) and Nb doped (001) STO substrates: (a) 100 nm reference film of SMO, (b) 20 nm SMO$_{0.5}$:BSO$_{0.5}$ VAN, and (c) 100 nm SMO$_{0.5}$:BSO$_{0.5}$ VAN.

**Structural investigations of SMO:BSO VAN film by XRD and HRTEM**. The epitaxial quality of three films were studied from XRD $2\theta$-$\omega$ scans, as shown in the Supplementary Fig. 1a. All the films show sharp (001) peaks, just to the left of the STO peaks. The peak at lower $2\theta$, labelled 'S' is understood by the fact that the SMO (structural information in Supplementary Table 1) is in-plane compressed by the STO (bulk average pseudo-cubic lattice parameter of SMO is 3.944 Å, and STO 3.905 Å) and hence out-of-plane tensed. For the 100 nm plain and VAN SMO films, an additional broad higher angle peak corresponding to relaxed SMO is observed. This corresponds to relaxed SMO, and is labelled 'R',

with $c$-axis lattice parameter of ~3.746 Å. The amount of relaxed SMO is large in the 100 nm plain film, as would be expected for a standard film of this thickness well above the critical thickness. It is only minor in the 100 nm VAN film and is relaxed to a much lesser extent (as observed from the strong overlap with the STO peak). This is because in VAN films the vertical strain state is controlled by domain matching epitaxy between SMO and BSO. This is discussed in more detail later. The relaxed peak is not observed for the 20 nm VAN film since this film is thin enough for the strain to be dominated by the substrate.

From X-ray $\varphi$-scans (Supplementary Fig. 1b) the films are highly aligned in-plane (predominantly 45° rotated in-plane) with an epitaxial relationship of [100]SMO//[110]STO or [010]SMO//[110]STO. This is expected as SMO has the GdFeO$_3$ structure ($\sqrt{2}a_p \times \sqrt{2}a_p \times 2c$), where $a_p$ and $c$ are, respectively, the in-plane and out-of-plane unit cell lattice parameters of a simple tetragonal perovskite unit cell.

As we show later, the 100 nm thick VAN SMO:BSO film shows the most interesting FM and FE properties of the films studied. Therefore, we focus here on the nanostructure, phase composition, and phase distribution in the 100 nm VAN film. Scanning transmission electron microscopy (STEM) high-angle annular dark-field (HAADF) images, both in cross-section (Fig. 1a, b) and plan-view (Fig. 1d, e), as well as STEM energy dispersive X-ray spectroscopy (EDS) maps (Fig. 1c, f) show a clear phase separation between high-quality epitaxial SMO and BSO. The BSO is highly faceted with cubic facets as expected for the (001) orientation of this phase[5,15].

EDS maps and elemental line profiles of the cross-section and plan-view images (Fig. 1c, f) show no measurable Bi in the SMO phase and a ~1:1 Bi:Sm ratio in the BSO. The phase boundaries between the two phases in the VAN are very clean, i.e. no secondary phases are present, as expected, since the structure forms by self-assembly.

**Room temperature ferroelectric properties of VAN films.** Strong RT FE properties were observed in the 100 nm SMO:BSO

VAN films (Fig. 2). PFM measurements, amplitude and phase of piezoresponse as a function of bias voltage at RT are shown in Fig. 2a. The RT FE behaviour is very different from bulk SMO, which is not FE[13,27]. We recall that the ground state of bulk SMO is A-AFM and PE[13,27,28]. In contrast, our plain 100 nm SMO film also does not show any RT FE properties. The piezoresponse amplitude measured ($d_{33}$) of 6.7 pm V$^{-1}$ is as high as bismuth manganite (BiMnO$_3$) thin films[29]. The box-in-box phase mapping was measured in a 6 μm × 6 μm × 6 μm area, after polarising the film with the DC voltage from +5 V to −5 V to +5 V. A characteristic FE hysteretic behaviour is observed. The phase contrast for the opposite voltage (±5 V) remains stable after 24 h (Fig. 2b). The long retention time also confirms the stable FE behaviour of the VAN film. FE polarisation switching is observed by using PUND pulse tests (Fig. 2c).

PUND measurements allow the intrinsic polarisation to be determined, since these measurements mitigate any parasitic or leakage in the films by measuring remanent polarisation by subtracting the non-switching polarisation from the switching polarisation. The PUND pulses used were 500 kV cm$^{-1}$, with a 1 ms pulse width and 1000 ms pulse delays (Additional 0.1 ms and 0.01 ms data are shown Supplementary Fig. 4), allowing both switching (*) and non-switching (^) polarisations to be measured. The maximum net switching polarisation ($2P_R$) of 3.9 μC cm$^{-2}$ was evaluated using the relation: $2P_R = (\pm P^*) - (\pm P^\Lambda)$. The measured value of net polarisation is enhanced compared to other FE $o$-RMnO$_3$ films which typically have $P < 0.5$ μC cm$^{-2}$ and at a much lower temperature ($T_{C,FE} < 50$ K)[2,4–9]. The PUND measurement was also carried out locally on a ~200 nm area by the nano-PUND technique (Supplementary Fig. 5)[30,31]. Further FE hysteresis loops were measured for all three films. Only the thick (100 nm) VAN film shows FE hysteresis loops (Supplementary Fig. 3), confirming the PUND data. The PFM phase loop, PFM phase switching, FE hysteresis loops and two different PUND measurements together demonstrate the strong RT FE properties in our VAN films.

To investigate the point group symmetry of SMO in the VAN films, we carried out optical SHG measurement at RT. Figure 2d

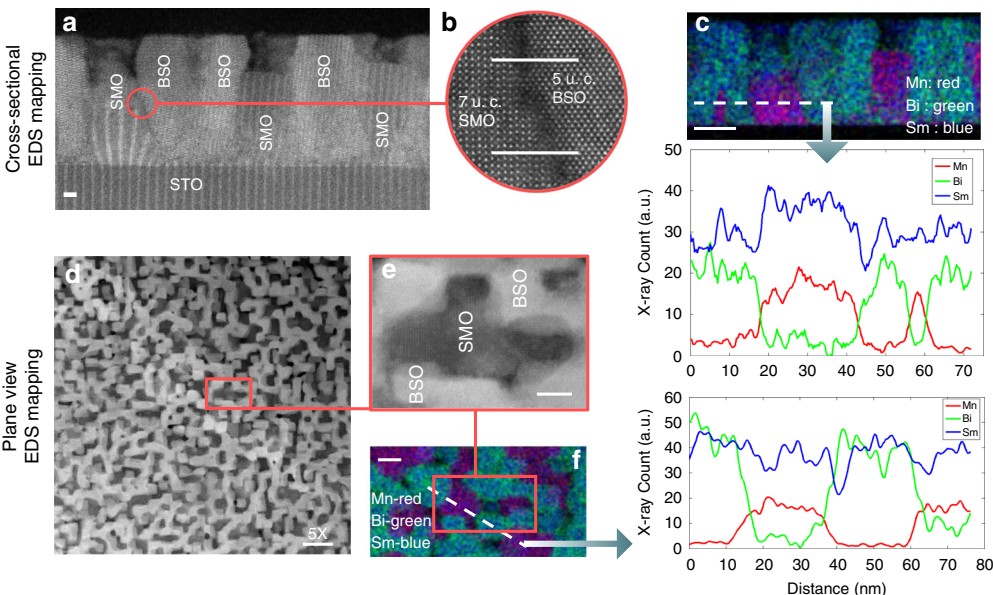

**Fig. 1 Electron microscopy images of thick SMO:BSO VAN film grown on STO (001). a** Cross-sectional STEM-HAADF image showing clear pillars of BSO embedded in SMO. The BSO grows faster than SMO as observed by the pillars being thicker than the SMO. **b** BSO and SMO showing domain matching epitaxy of 7 unit cells of SMO matching with 5 unit cells of BSO. **d** Plan-view STEM-HAADF image shows a VAN structure over a wide area. A regular pattern of connected BSO pillars in an SMO matrix is observed. **e** A region of SMO enclosed by laterally connected BSO pillars is shown. **c, f** EDS maps and line profiles of across the VAN film, showing that SMO contains only very minor Bi and that BSO has a ~1:1 Bi:Sm ratio. Scale bar: 10 nm.

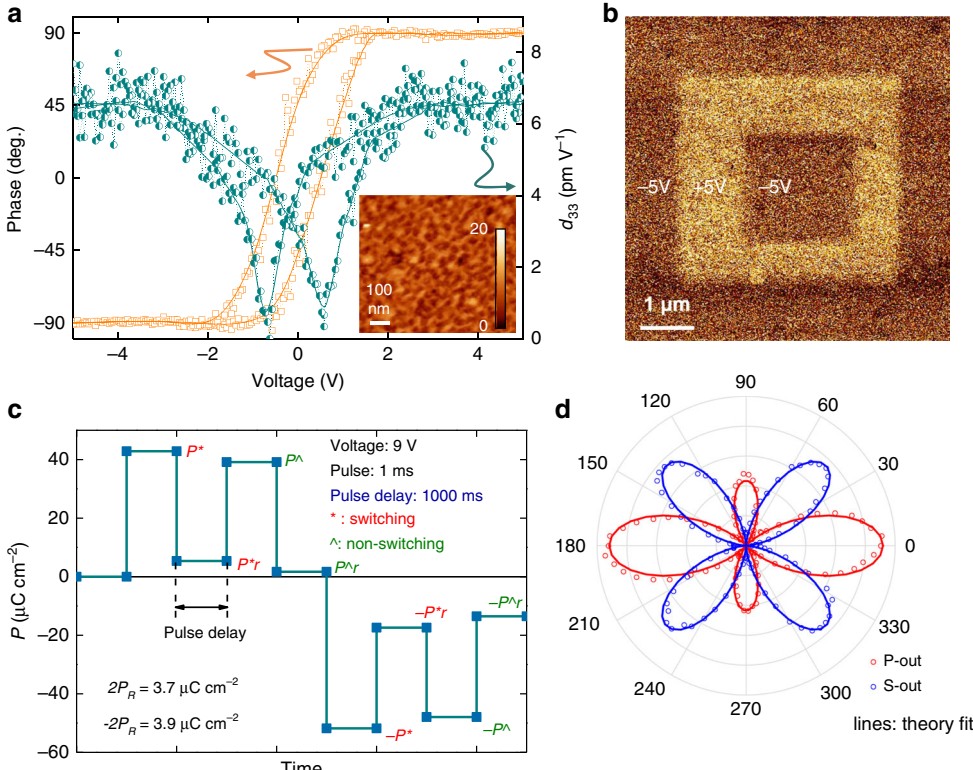

**Fig. 2 Intrinsic ferroelectricity for ~100 nm thick VAN SMO:BSO films at RT. a** Piezo-hysteresis response measured by PFM. Amplitude (circle and blue) and phase (square and red) of the PFM signal as a function of bias voltage. The inset AFM image shows the surface morphology of the film. **b** PFM phase contrast after −5 V writing (6 × 6 μm²), 5 V rewriting (4 × 4 μm²) and −5 V rewriting (2 × 2 μm²) at RT. The phase contrast remains after 24 h. **c** PUND measurements with 9 V ($E = 500$ kV cm$^{-1}$), 1 ms pulse width and 1000 ms pulse delays for capturing both switching (*) and non-switching (^) polarisations. The net switching polarisation, $2P_R$, is ≈3.9 μC cm$^{-2}$ and is consistent with $P_R$ obtained from $P$-$E$ hysteresis. **d** Optical SHG signals at RT. Red and blue points represent p-wave and s-wave of the second harmonic electric field. The incidence angle is 45°. The solid line is a theoretical fit of the SHG data.

shows the result of far-field reflection SHG polarimetry of s- and p-wave, at sample's tilt angle of 45° and in-plane angle of 0 or 90°. (See Supplementary Discussion on SHG). At the tilt angle of 0°, no SHG signal was detected, indicating that the tetragonal c-axis points out of the film's surface. The theory fit to the experimental SHG polar plots (lines in Fig. 2d) shows that the macroscopic pseudosymmetry point group of 100 nm SMO VAN film is identified as the polar point group of 4 mm. This result is in a good agreement with the X-ray φ-scans data. The temperature dependent SHG up to 623 K (350 °C) is conducted with 45° reflection p-in, p-out geometry (Supplementary Fig. 6). The SHG signal decreases with increase of temperature indicating that there is a non-centrosymmetric to centrosymmetric structure transition over a broad temperature range. The change in slope showing onset of saturation of SHG to be below ~360 K–370 K. The origin of the second hump in SHG near ~500 K is currently of unknown origin.

**Magnetic properties of VAN films.** The magnetic properties of the 20 nm and 100 nm-thick VAN films are compared in magnetisation versus temperature $M(T)$ plots in Fig. 3. The $T_{C,FM}$ are determined to be ~70 K and ~90 K for the 20 nm and 100 nm films, respectively. In both films, a possible cluster-glass like behaviour is observed at ~20 K where the magnetic moment or susceptibility reach maximum values[32,33]. This is likely because of spin canting arising from the competition between AFM and FM couplings as is commonly observed in BiMnO₃[34]. The inset of Fig. 3 compares magnetic hysteresis (MH) loops for these films at 10 K. Both films are FM with a coercivity ($H_C$) of ~40 mT at 10 K. The saturation magnetisation ($M_S$) values at ±1 T are 30 emu cc$^{-1}$

and 100 emu cc$^{-1}$, respectively. The drastic increase in magnetisation in the thicker film is a prominent indication of a significantly modified spin structure in the upper part of the thicker film. The lack of magnetisation saturation in both films, even above 1 T, is a further indication of cluster-like glass behaviour.

**Discussion**

To understand more about why the 100 nm VAN film shows the unusual RT FE behaviour at the same time as strong FM, a more detailed analysis of the crystal structures of the three different films was undertaken by high-resolution asymmetric X-ray reciprocal space maps (RSMs) around the (113) reflection of STO. For all the films, the region around the (113) reflection of STO revealed split (206) and (026) peaks of SMO indicative of the orthorhombic SMO structure[5,15,35–37] as expected from the bulk structure of SMO (see Supplementary Table 1). We label the orthorhombic phase as o-SMO1, with the peak position overlapping strongly with (113) STO owing to the coherent growth on STO. This o-SMO1 phase corresponds to 'S', the strained phase labelled in Supplementary Fig. 1a. It is noted that split peaks from the orthorhombic structure are not observed in Supplementary Fig. 1a because of the close positions of (206) and (026) peaks.

A clear difference in the RSMs of Fig. 4 emerges for the 100 nm VAN film cf. the 20 nm VAN film and the 100 nm plain film. A new split peak appears at larger $Q_X$ (smaller in-plane lattice parameter) and lower $Q_Z$ (larger out-of-plane lattice parameter), than for o-SMO1. This peak is labelled as o-SMO2. The splitting of the peaks again indicates an orthorhombic structure.

To understand the origin and evolution of the o-SMO2 phase, and to understand why the VAN structure promotes its

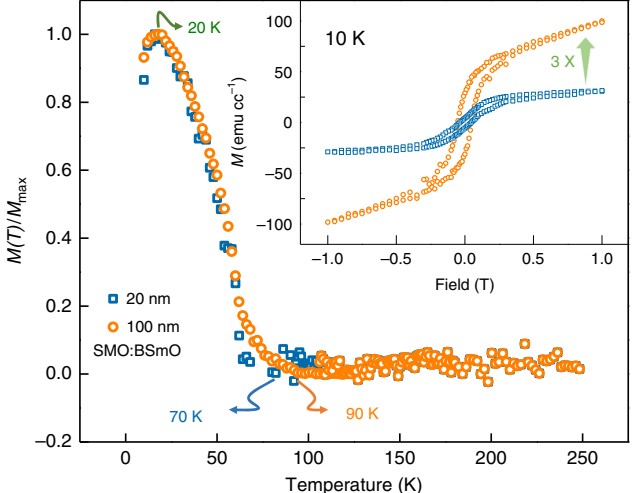

**Fig. 3 Magnetic properties of 20 nm and 100 nm thick VAN SMO:BSO films.** Main panel shows normalised in-plane field cooled $M(T)$ at a field of 20 mT. The $T_{C,FM}$ values are ~70 K and ~90 K for the 20 nm and 100 nm thick VAN films, respectively. The inset shows $MH$ loops for the 20 nm and 100 nm films, respectively, at 10 K, measured with field range ±1 T, with the field applied in-plane. The $MH$ loops show that the volume magnetisation increases as the thickness increases.

formation in the thicker VAN film, the lattice parameters of the $o$-SMO1 and $o$-SMO2 phases were determined and compared. This was done by peak fitting of the RSM peaks as explained in the experimental section. The results are shown in Supplementary Table 1 where the overview of structural and physical properties of all films are compared. Two different misfit strain values are shown in Supplementary Table 1.

Top of table: 'Measured strain (%) in the SMO films (in the three orthogonal directions) $cf.$ bulk SMO (bulk STO)'. For example along the $a$ direction, the strain is $(a_{film} - a_{bulk})/a_{bulk} \times 100\%$. The strain with respect to bulk STO is calculated only for the 100 nm VAN film since the strain in this film was also calculated from STEM images, and we are interested in particular in how the strain changes through-thickness in this film.

Bottom of table: 'Calculated strain (%) in bulk SMO required to match $\sqrt{STO}$ (along the three orthogonal directions)'. For example, along the $a$ direction, the strain is $\sqrt{2}a_{STO} - a_{SMO})/\sqrt{2}a_{STO} \times 100$.

Looking first at the calculated strain in bulk SMO required to match STO (bottom half of Supplementary Table 1) there is an anisotropic misfit strain with STO because of the different $a$- and $b$-lattice parameters of SMO both in bulk and VAN. The strain equates to +3.0% along the $a$-axis and −4.7% along the $b$-axis, or an average in-plane strain of −1.0%. Hence, plain SMO films on STO will be compressed in-plane by STO and tensed out-of-plane. This was confirmed in Supplementary Fig. 1a.

Looking now at the strain in the three different films, we see that in the $o$-SMO1 phase the in-plane strain levels (2.7–2.8%) are very close to the calculated values for bulk SMO strained to STO (3%). Hence, for all the films, $o$-SMO1 is coherently strained to the STO. This occurs because $o$-SMO1 is in the bottom part of the film where SMO is adjacent to the STO substrate surface. The out-of-plane strain levels (i.e. along $c$) (>4% w.r.t. bulk SMO) are relatively high for all the films and this is likely related to non-stoichiometry effects which are common to manganite films.

In the 100 nm VAN film (which is of most interest to us here because this film shows RT FE behaviour), we observe that the $o$-SMO2 phase has a very different strain state to either the $o$-SMO1 phase or to bulk SMO. Strain values w.r.t. bulk SMO of −0.4% along $a$, and −6.3% along $b$ are observed. Along $d_{110}$ this equates to

−3.6% (or −2.5% w.r.t. STO). Notably, the average in-plane strain is much larger in $o$-SMO2 than in $o$-SMO1 where it is only ~−0.9% w.r.t. SMO, or 0.01% w.r.t. STO. This is because the $o$-SMO1 is coherently strained by the STO, as already mentioned.

Along the $c$ axis, the strain in $o$-SMO2 is 4.9% w.r.t bulk SMO, or 0.64% w.r.t. STO. Normally, these high strain levels are not maintained in multiferroic films as misfit strain is reduced by nanoscale twin domain structures[5,15]. It is likely that relaxation is hindered in the VAN films because the critical twin domain size is larger than the nanopillar size (which is 10's of nm in our films (Fig. 1)).

In the 100 nm VAN films, the in-plane and out-of-plane strain values determined from the X-ray data (discussed above) were fully confirmed by lattice strain calculations from high-resolution STEM. A continuously increasing in-plane compressive strain up to ~5.0% w.r.t bulk SMO (0.64% w.r.t. STO) was measured from STEM images. Details of these measurements and calculations are shown in the Supplementary Fig. 2 and accompanying discussion.

The origin of the large in-plane compression in $o$-SMO2 is linked to how the BSO nanopillars strain the SMO in-plane. In plain films, only the substrate influences the SMO lattice parameters whereas in the VAN films, it is well known that the pillars can strongly influence both the vertical and in-plane strain states, and with increasing effect with film thickness[20,38].

During growth the BSO nanopillars give a nano-pressure-chamber effect because of the relatively faster growth of BSO compared to SMO. The faster growth rate is confirmed by the BSO nanopillars being taller than SMO in the HR-TEM image of Fig. 1a, and also by the fact that the pillars are connected together in-plane to form a maze-like structure, as shown in Fig. 1d. Hence, they must overgrow the SMO, impinging on it, and squeezing it in-plane. Also, upon cooling the pillar shrinkage can lead to a further in-plane compression of the SMO matrix[21]. This is because the BSO is stiffer than the SMO[21,23–26].

A schematic of the 3D strain effect in the VAN films, showing how it emerges with thickness is shown in Fig. 5. Figure 5a–c illustrates the thin VAN film, with Fig. 5a showing a 3D sketch of the film microstructure, Fig. 5b showing that the SMO is 45° rotated in-plane, and Fig. 5c showing a schematic of the crystal structure matching in a film cross-section. We label the thin VAN film as being in the 'a-region', corresponding to the $o$-SMO1 phase. The thick VAN film is shown in Fig. 5d–f. Now, two regions are identified corresponding to $o$-SMO1 ('a-region') at the bottom of the film and $o$-SMO2 ('b-region') in the upper part of the film. There is an intermediate region where a transition of $o$-SMO1 to $o$-SMO2 takes place. In the 'b-region' the enclosing BSO nanopillars control the strain in the film, giving the large in-plane compression and out-of-plane tension in the SMO pillars.

The origin of the moderately enhanced vertical strain in $o$-SMO2 compared to $o$-SMO1 (4.9% w.r.t. bulk SMO $cf.$ 4.1%) is the vertical domain matching epitaxy of the SMO with BSO. The vertical domain matching of BSO to SMO is 5 unit cells along $c$ of BSO match to 7 unit cells of SMO, as shown in Fig. 1b. Considering that 5×BSO:7×SMO = 5×11.1017 (=55.5085 Å): 7×7.493 (54.2451 Å). Since 55.5085 Å > 54.2451 Å, the SMO is estimated to be stretched by ~2% at the interface with BSO. In fact, we observe an extension of just under 1% more than that in the plain film. It is likely less than calculated because the strain is also accommodated by stoichiometry modification.

Overall, there is a decrease of the cell volume in $o$-SMO2 compared to $o$-SMO1 (240.7 Å$^3$, $cf.$ 227.7 Å$^3$). This is mainly dominated by the decrease of $a/b$ lattice parameters ($d_{110}$ strain −3.6%). Hence the Mn–O bond length will be decreased in line with the in-plane compression and this will modify the in-plane magnetic interactions of Mn moments and spin-order states[10,14,39]. On going from $o$-SMO1 to $o$-SMO2, since the $a$ and $b$ lattice parameters shrink but $c$ expands, the Mn–O–Mn

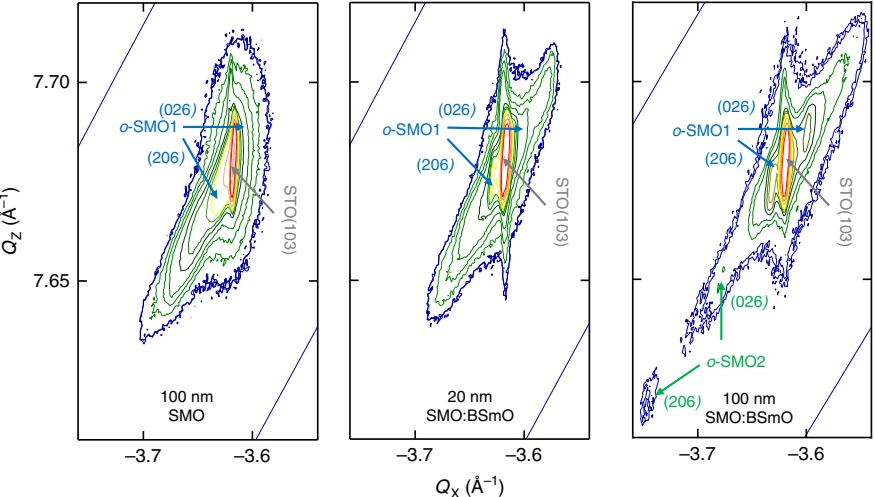

**Fig. 4 Structural analysis.** Asymmetric RSMs around the (113) reflection of STO. (206) and (026) reflections of orthorhombic SMO (o-SMO1) appear clearly in the RSMs of three different films confirming the presence of the o-SMO1 orthorhombic crystal domain structure. For the 100 nm SMO film, the orthorhombicity is less than in the SMO:BSO VAN films (as indicated by less splitting of the o-SMO1 peaks). The 100 nm SMO:BSO VAN film shows an additional orthorhombic phase (o-SMO2) indicated by the split peaks at lower $Q_Z$ and higher $Q_X$ than the STO peak position.

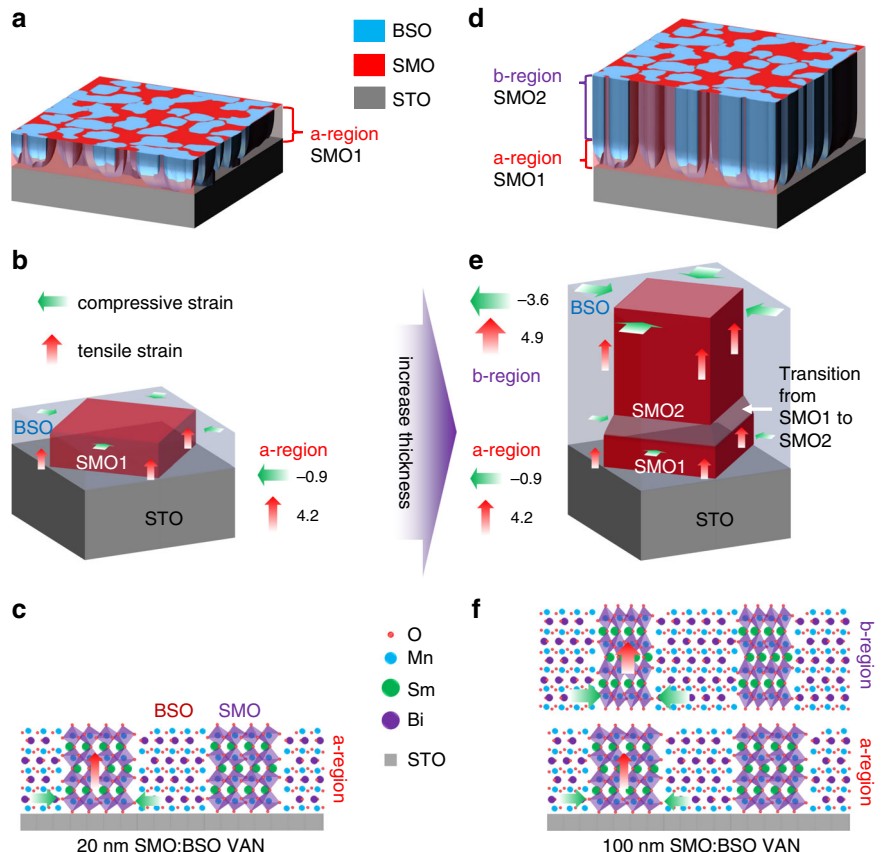

**Fig. 5 Model of structure and strain evolution in SMO VAN Films. a–c** In the 20 nm VAN film, SMO is clamped to the STO substrate. This takes place by '$a$' expanding while '$b$' contracts. '$c$' expands marginally. This produces the o-SMO1 phase i.e. the 'a-region'. **d–f** As the film thickness increases to 100 nm, the influence of the STO is much weaker and now vertical strain from the BSO pillars dominates the strain state in the film. Here, '$c$' expands owing to domain matching epitaxy growth vertically with BSO. At the same time, there is biaxial lateral (in-plane) pressure from the BSO on SMO leading to a large in-plane contraction along both '$a$' and '$b$'. Hence, the compressive in-plane ($d_{110}$) strain (green arrow) and tensile out-of-plane strain (red arrow) increases from 4.2% and −0.9% in o-SMO1 to 4.9% and −3.6% in o-SMO2 respectively.

out-of-plane bond angle increases and the in-plane Mn–O bond length decreases.

We recall that there was a ~3 times stronger volume magnetisation signal in the 100 nm VAN film (containing o-SMO1 and o-SMO2) cf. the 20 nm VAN film (containing o-SMO1 and possibly some o-SMO2, the latter too weak to be observed in XRD). This proves that o-SMO2 is FM (Fig. 3). We now consider whether the o-SMO1 phase in the VAN films could be weakly FM and FE. Considering that, the strain levels of o-SMO1 in the 20 nm and 100 nm VAN films are very similar to the plain SMO films (Supplementary Table 1) which are non-magnetic and PE, it is likely that o-SMO1 has the same non-magnetic and PE properties. From the volume magnetisation values, the critical thickness of o-SMO1 is estimated to be ~15 nm and so the top ~5 nm of the 20 nm VAN film would be the o-SMO2. This top layer would give the FM observed for the 20 nm VAN film in Fig. 3. The reason that no FE was measured in this top layer is because it would be too thin for the FE to be measured.

To quantitatively understand the effect of the strain on the Mn–O bond length and Mn–O–Mn bond angles and hence to explain the origin of the ferroelectricity in o-SMO2, DFT calculations were performed for the o-SMO1 and o-SMO2 structures. Fixed-cell geometry optimisations were carried out using experimentally measured lattice parameters tabulated in Supplementary Table 2. It is known that DFT systematically underestimates/overestimates the lattice constants but only a relative comparison here is needed in order to reveal the effect of strain on the ionic co-ordinates. The calculated bond lengths (Supplementary Table 2), show a significant change of the crystal structure in o-SMO2 compared to o-SMO1.

For o-SMO1, the average Mn–O bond in-plane bond length is calculated to be 2.038 Å. However, for o-SMO2 the calculated value is lower, at 1.984 Å, i.e. by −2.65%. This is in close agreement with the experimentally measured difference from Supplementary Table 1. The calculations also show that the out-of-plane Mn–O–Mn bond angles increase moderately from 143.722° in o-SMO1 to 147.361° in o-SMO2 (i.e. a change of ~3.6°). These changes in bond lengths and angles are much higher than those achieved by physical pressure methods previously (e.g. 0.8° and 1.1%, respectively, at 10 GPa in TbMnO$_3$ which has a highest reported $P$ of all $R$MnO$_3$ (at 5 K))[18].

In $R$MnO$_3$, it is well known that the bond angles and lengths (modified by changing $R$ to induce chemical pressure or by using physical pressure[16]) are critical to the magnetic properties. The magnetic spin ordering is changed from A-AFM to E-AFM when the temperature is reduced or the structural distortion increases, causing $J_1$ and $J_2$ to be increased[18]. The large pseudocubic strain of −3.6 % in o-SMO2 is expected to destroy the A-AFM ordering present in bulk SMO, changing it to E-AFM[17,18]. Furthermore, theoretical calculations show that under a large compressive strain of ~−4 %[17] the E-AFM structure is not stable and FM ordering emerges with highly enhanced polarisation. The predicted FM ordering temperature ($T_{C,FM}$) is ~80 K[17].

Here, the reduction of the Mn–O bond length and the increase of the Mn–O–Mn bond angle of o-SMO2 are consistent with the proposed mechanism of Mn–Mn interactions with both $J_1$ and $J_2$ being enhanced. These enhanced interactions explain the ferromagnetism and the RT FE polarisation in the 100 nm thick SMO: BSO VAN film. Hence, our experimental results for the o-SMO2 phase match the theory very well both for the level of in-plane strain (−3.6%) and for the $T_{C,FM}$ ~90 K and enhanced FE polarisation.

Moreover, under in-plane compressive strain, the staggered $d_{3x^2-r^2}/d_{3y^2-r^2}$ type orbital ordering in $R$MnO$_3$ which has a small GdFeO$_3$-type distortion (with A-AFM)[13] changes to a mixture with $d_{x^2-z^2}/d_{y^2-z^2}$ states[17]. This enhances the asymmetric hopping of e$_g$ electrons between the Mn and subsequently enhances the

net FE polarisation[17]. Therefore, in o-SMO2 (in the 100 nm VAN SMO:BSO film) the main origin of enhanced ferromagnetism and RT FE polarisation in 100 nm thick SMO:BSO is 3D strain exerted by the BSO nanopillars.

On a final note, it is possible that further tuning of strain in o-$R$MnO$_3$ phases using different VAN compositions could yield even higher temperature multiferroicity. Also, it is possible that investigating 'asymmetric hopping' of Mn-e$_g$ electrons in VAN structures containing other multiferroic materials, e.g. BiFeO$_3$ films with mixture of R-and T-phase on STO or the h-$R$MnO$_3$ structure, could lead to RT multiferroicity.

In this study, by 3D strain engineering of SmMnO$_3$ in nanocomposite films, we report spin-driven ferroelectricity at room temperature with clear high $T_{C,FM}$ ferromagnetic behaviour. This compares to bulk SmMnO$_3$, which has a ground state of paraelectric and A-type antiferromagnetism with a $T_N$ of ~ 60 K. The net switching polarisation (2$P_R$) and piezoresponse amplitude ($d_{33}$) are 3.9 μC cm$^{-2}$ and 6.7 pm V$^{-1}$, respectively. This compares with previous reports of GdMnO$_3$ films[5] being with FE $T_{C,FE}$ of 75 K and TbMnO$_3$ films having electric polarisation of ~1.8 μC cm$^{-2}$ (but at very low temperatures 5 K, external high pressures 5.2 GPa and high magnetic field 8T)[18]. In addition to the enhanced ferroelectricity, the ferromagnetic transition temperature ($T_{C,FM}$) and saturation moment ($M_S$) of SmMnO$_3$ were ~90 K and 1.02 μ$_B$ Mn$^{-1}$ at 10 K, respectively. The enhanced ferroelectricity of SmMnO$_3$ was only present in the thicker (100 nm) nanocomposite films where the in-plane compression is much larger than in the thinner films. As determined from DFT calculations, the large in-plane compression leads to the change of Mn–O bond angle and length, which indicates an enhanced exchange interaction, consistent with the experimentally observed ferromagnetism and room temperature ferroelectricity.

## Methods

**Sample preparation.** Self-assembled VAN thin films of (SmMnO$_3$)$_{0.5}$:((Bi, Sm)$_2$O$_3$)$_{0.5}$ were grown on both (001) SrTiO$_3$ (STO) and Nb-doped SrTiO$_3$ (Nb: STO) substrates by pulsed laser deposition (PLD). Film thicknesses of 20 nm and 100 nm were grown. As a reference, a SmMnO$_3$ film of 100 nm thickness was also grown. The targets were prepared by mixing appropriate ratios of high purity Bi$_2$O$_3$, Mn$_2$O$_3$ and Sm$_2$O$_3$ powders with 10% Bi excess to give the Bi:Sm ratio of 1:1 in the (Bi,Sm)$_2$O$_3$. The targets were sintered from 900 °C (VAN films) – 1000 °C (plain SMO). A deposition temperature of 650 °C was used for all films. The laser pulse rate was 2 Hz and the laser fluence was 1.5 J cm$^{-2}$. The oxygen pressure was fixed at 100 mTorr during the deposition.

**XRD analysis.** To confirm the phase and the crystalline quality of the thin films, detail high-resolution 2θ-ω XRD scans were carried out on a Panalytical Empyrean high-resolution XRD system at room temperature. To explore the 3D strain state, asymmetric RSMs around (113) of STO were also carried out. The a and b lattice parameters were determined by peak fitting of RSM scans using the Epitaxy® software package. For the o-SMO1 phase, the c parameters were also calculated from 2 theta-omega scans, were cross-checked against the RSM values and were found to be the same within ±0.01 Å. Details of structural information can be found in the Supplementary Discussion on XRD.

**HRTEM analysis.** Detailed structural properties of the films were investigated by HRTEM (JEOL 2010 microscope) operating at 200 kV and a JEOL 4000 EX microscope operating at 400 kV) and a FEI TitanTM G2 80-200 STEM, with a Cs probe corrector, operated at 200 kV. EDS was used for the element distribution mapping.

**Magnetic measurements.** Detail magnetic measurements were carried out using a superconducting quantum interference device (SQUID) magnetometer (Quantum Design, MPMS). The magnetic moment was converted from emu cc$^{-1}$ to μ$_B$ Mn$^{-1}$ by using the unit cell volume obtained from XRD.

**Ferroelectric measurements.** Atomic force microscopy was used to determine the surface structure of the films. PFM measurements were performed using an Agilent 5500 Scanning Probe Microscope (Agilent SPM 5500) in a PFM mode with 3 (MAC-3) lock-in amplifiers (LIAs) at RT. For the PFM measurements, an Olympus Pt-coated tip (Asylum Research AC240TM) was used at an excitation frequency of

15 kHz with an alternating current (AC) voltage VAC of 2 V, polarised by a direct current (DC) bias of VDC of ±5 V. The inverse (also called converse) piezoelectric effect was used to induce longitudinal (thickness) film displacements on a local scale and beneath the Pt surface electrodes (of ~250 μm in diameter). The PFM displacement was influenced little by the electrostatic force between the sample and the cantilever assembly, since they were at the same electrical (whole cantilever body was electrically screened by the tip) and contact (Pt-tip and Pt-patch) potential.

To characterise the FE properties, Polarization—Electric field (P–E) hysteresis loops were measured using a Radiant precision LC analyzer at room temperature. Further switching of polarisation was tested by using the PUND pulse method. In the PUND measurements, a series of 5 pulses were applied at 300 kV cm$^{-1}$ with pulse sequence of different pulse times (1 msec, 0.1 msec and 0.01 msec) to capture both switching and non-switching polarisations. The required switching electric field of 500 kV cm$^{-1}$ was relatively low (e.g. 9 V applied on the SMO$_{0.5}$:BSO$_{0.5}$ nanocomposite films of 100 nm thickness). A delay of 1000 ms was set between the pulses. The initial pulse was applied to preset the film to the polarisation stage and no measurement was made at this point. A second pulse (Pulse 1) was applied to switch the sign of polarisation. After a 1000 ms pulse elapse, the film was left to relax, allowing the non-remnant polarisation to be dissipated. A third pulse (Pulse 2) was then applied and the polarisation was measured without pre-switching. Fourth and fifth pulses (Pulse 3 and 4) were then applied, similar to Pulse 1 and 2 but with the opposite polarity. To calculate the maximum net switching polarisation (2P$_R$) the full area of the electrode was considered, Since the VAN film is made up of two phases, the calculated polarisation value represents a lower bound. The non-FE BSO phase forms around half the volume of the film and so the polarisation values could be up to around two times larger than has been estimated.

PUND measurements at the nanoscale were also conducted with PFM using the nano-PUND (also termed AFM-PUND) method[30,31]. The measurements were done by applying switching and non-switching voltages—via the back (substrate) electrode of the sample and a conductive AFM tip connected to a trans-impedance amplifier with the input at virtual zero. In this method, the triangular sweeps were applied with either positive and/or negative voltage, in pairs.

**SHG experimental setup.** Far-field Reflection SHG polarimetry was performed with 800 nm fundamental laser beam generated from Empower 45 Nd:YLF Pumped Solstice Ace Ti:Sapphire femtosecond laser system (pulse width of 95 fs and repetition rate of 1 kHz). The schematic figure for the far-field reflection SHG is shown in Supplementary Fig. 7. The 800 nm laser beam is radiated onto the thin film sample with the sample tilt angle of θ (0 and 45°). The tilt angle θ is defined as the angle between the sample normal and the wavevector of incident beam. The in-plane orientation of the sample is defined by ψ (0 or 90°). The polarisation state of the light is defined by angle φ. φ is rotated from 0° to 360° by λ/2 (half-lambda) wave plate. The second harmonic electric field (400 nm) reflected from the sample is separated into vertical (s$_{out}$) and horizontal (p$_{out}$) components by polarised beam-splitter, then the SHG intensity of each component is measured by a photo-multiplier tube (PMT). To ensure that there is no SHG signal from the substrate, the SHG signal from substrate was removed by defocusing the fundamental beam.

**Density functional theory.** The plane wave pseudopotential code CASTEP was used for DFT calculations. A 6×6×6 Monkhorst-Pack grid for k-points and plane wave cut-off energy of 700 eV were used. The PBEsol exchange correlation functional was used since it gives equilibrium lattice constants close to experimentally measured values. The pseudopotentials were used to treat the valence electrons for 2s 2p states in O, 3s 3p 3d 4s states in Mn and 5s 5p 6s 5d state in Sm. The A-type AFM arrangement of spins was used for Mn atoms. A value of $U = 4$ eV was used to correct the self-interaction error for $d$ electrons in Mn. The 4f states in Sm were included in the core, as otherwise, they could cause instabilities during self-consistent cycles. Since we are only interested in the change in bond angle and bond length with applied strain, neither the choice of $U$ value nor the omission of Sm 4f electrons would significantly affect the results.

## Data availability

All the experimental/calculation data that support the findings of this study are available from the corresponding authors upon reasonable request. All the codes used for this study are available from the corresponding authors upon reasonable request.

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

## Acknowledgements

The investigators in Cambridge acknowledge support from EPSRC grants EP/L011700/1 and EP/N004272/1, and the Isaac Newton Trust (Minute 13.38(k) and RG96474). T.M. and J.L.M.D. also acknowledge funding from EU grant H2020-MSCA-IF-2016 (745886)-MuStMAM. J.L.M.D. also acknowledges funding from the Royal Academy of Engineering, CiET1819_24. B.Z. acknowledges support from China Scholarship Council and Cambridge Commonwealth, European and International Trust. Sandia National Laboratories is a multi-programme laboratory managed and operated by National Technology and Engineering Solutions of Sandia, LLC., a wholly owned subsidiary of Honeywell International, Inc., for the U.S. Department of Energy's National Nuclear Security Administration under contract DE-NA0003525. Y.P. and V.G. acknowledge support from the Department of Energy grant, DE-SC0012375. S.Y. was supported as part of the Computational Materials Sciences Program funded by the US Department of Energy, Office of Science, Basic Energy Sciences, under Award Number DE-SC0020145.

## Author contributions

E.C. and T.M. contributed equally. E.C., T.M. and J.L.M.-D interpreted the data and wrote the paper with help of other authors. E.C. and TM did sample preparation, magnetic and ferroelectric PUND measurements. E.C. carried out XRD measurements. A.K. performed PFM and AFM-PUND measurements. R.W. took part in manuscript preparation. B.Z. did DFT calculations. P.L., Z.B. and H.W. did TEM analysis. S.Y., Y.P. and V.G. did SHG analysis.

## Competing interests

The authors declare no competing interests.
