## [Peer Review File · Nature Communications]

Reviewers' comments:

Reviewer #1 (Remarks to the Author):

This manuscript describes the study of strain-induced property evolution in nanocomposites for SmMnO₃ with (Bi, Sm)₂O₃ nanopillars embedded in the SmMnO₃. As noted, materials like SmMnO₃ are known to be type-II multiferroics, but generally have ordering temperatures that are well below room temperature. Here, the production of large in-plane compressive and out-of-plane tensile strains are said to produce room-temperature polarization in the SmMnO₃. The magnetic order is found to be ferromagnetic with a critical temperature of ~90K while bulk SmMnO₃ is antiferromagnetic. Finally, density functional theory is used to explore the mechanism giving rise to the ferroelectric order which is stated to arise from a "modification of the exchange coupling."

The main observations here are the large increase in the FE transition temperature (above RT vs. < 40 K in bulk), the large polarization (~4 uC/cm²), and the observation of FM order with a T_c ~ 90 K. At face value this seems quite dramatic. This said, there are a number of statements which could be better supported by experimental proof to confirm these claims. In particular, attention is drawn to measurements of ferroelectricity and the Mn-O-Mn bond angles. Both of these are considered lacking at this time and additional studies (described below) would be appropriate to improve these statements. As it stands, the paper has some speculative aspects at this time. Finally, the writing is disjointed, there are lots of small, short paragraphs. Overall it lacks the narrative form expected for a publication in a journal like Nature Communications. Thus, at this time, publication seems premature in a journal like Nature Communications.

Detailed comments:

- Call the ferroelectricity "giant" is misleading and probably not appropriate. To be clear, polarization of 3.9 uC/cm² is not "giant" compared to regular ferroelectrics. This comes off a bit aggressive. It is noted that it is quite a bit larger than that observed in bulk SMO, but again it is not giant. This should be changed and removed from the paper.
- The section (on page 2) concerning the vertically aligned nanocomposites (VANs) is awkward for two reasons. First, it comes across as bit like a "sales pitch" (if you will) – strongly suggesting this is the only answer. Second, the bulleted list is odd and not something this Reviewer has seen in a large number of publications. This Reviewer would prefer a more passive, factual statement of approaches that have been used – advantages/disadvantages are fine – but less like a sales pitch for this approach.
- The statement "The film compositions are assumed to be the same as the target compositions used" seems out of place and unnecessary. First, without measurement such an assumption is unfounded. Second, it is becoming more clear that such assumptions are difficult to make – even in this system where the materials spontaneously form. Also, the paper itself notes that this specific system was picked because if some Bi ended up in the SMO it wouldn't be a bad thing.
- The language (page 3) "the film will be out-of-plane tensed" is not clear in meaning or consistent with literature discussions on such a matter.
- Why are the peaks so broad, rounded for the film in Fig. S1b?
- Proof of ferroelectricity – The proof of ferroelectricity is some what questioned here. This is perhaps not a major fix, but the data provided seems an interesting choice. It is strongly recommended that the manuscript provide ferroelectric hysteresis loops for all samples – VANs and single films – and report the PUND data in a more traditional way – showing switching both as 1) a function of pulse time with different voltages and 2) as a function of voltage at different pulse times. The PUND data as shown is un processed and an odd choice. The Authors can find numerous examples of how such data is presented in the literature. The statement that SHG confirms ferroelectricity (page 3) is again misleading. SHG probes noncentrosymmetry/polar structures – switching is not shown in the SHG data. So the SHG technically proves polarity, but not ferroelectricity. Finally, the authors are asked to refrain from using PFM as proof of ferroelectricity. Together with the other measurements it can support such a statement, but alone it is hard to say. There are many ways to get contrast like that imaged here from PFM.

- Ferroelectric measurements – Please provide details on how the measurements were done. For the hysteresis loops, please provide frequency dependent studies and show that they follow expected trends. The biggest question here is how the area was determined considering that the material is not homogenous either in-plane or out-of-plane, thus some error bars on the value of polarization seem essential.
- SHG – It is stated that SHG was done to extract the point group symmetry of the VANs, but that point group symmetry is not given. A point group of 4 is provided – is this the full symmetry? No other symmetry elements? This symmetry needs to be self-consistent with the proposed structural changes later, is it? Second, SHG provides an excellent way to extract the ferroelectric transition temperature – this not done, but should be to further confirm the attribution of ferroelectric order. Switching and a transition temperature will go a long way to show this.
- Are the authors sure that the elongation seen in Figure 4 is splitting of a peak and not either measurement artifact, partial relaxation, or some combination thereof? Where should the strained and relaxed peak positions be for the o-SMO in these scans? Can this be marked?
- The schematics in Figure 5 (and the associated text) are a bit hard to follow. There is a big leap of faith to be made to believe this is the “full” picture. Can direct measurements in the STEM be done to prove this? Furthermore, if the structure/strain is varying throughout, can we be sure about the extracted/reported values of magnetism? Polarization? Aren't they likely to also be changing?
- The assertion that the Mn-O-Mn bond angle is not based on a measurement, but is deduced from the structural changes. As such it is a bit speculative. Direct proof of this would be nice.

Reviewer #2 (Remarks to the Author):

Obtaining large and high-temperature magnetization and polarization in multiferroics is still a major challenge. Working with composite materials may help to avoid this dilemma, but because of their grainy, irregular structure any magnetoelectric coupling in these is usually mediated by periodic deformation and therefore an AC effect occurring under specific circumstances only. With their work, the authors bring together the advantages of crystalline and composite multiferroics. They grow a SmMnO₃/(Bi, Sm)₂O₃ nanopillar composite, yet as epitaxial film with well-defined interfaces between the constituents. They thus obtain above-room temperature ferroelectricity with unusually high spontaneous polarization and a sizable magnetization up to almost 100 K. Key to these pronounced multiferroic properties are the high strain values they can stabilize by the nanopillar architecture.

This work demonstrates once more that the toolbox for obtaining pronounced multiferroic responses towards higher temperature is not yet empty. I find the approach of using epitaxial nanocomposites original and useful and the performance parameters obtained with it remarkable. To my opinion, this work deserves publication in Nature Communications.

There are a couple of open issues the authors should work on prior to acceptance.

(1) My main point is that I do not quite understand the relation between ferromagnetic order and the cluster-glass-like behavior. From the text, I am not sure in what temperature range exactly is the cluster-glass-like response observed. So how does it interfere with the notion of ferromagnetism: Can one talk about ferromagnetism at all if a cluster-glass state is present?

(2) What is the ferroelectric Curie temperature? As the spontaneous polarization is detected by second harmonic generation (SHG), above-room-temperature measurements up to T(C) should be possible because thermal conductance is not detrimental to SHG.

(3) Is it possible that the polarization is caused by interface charges rather than by the onset of ferroelectricity? Interface charges may be switchable, too, and are not necessarily detected by a PUND measurement.

(4) The point symmetry of the film is concluded from the SHG measurements as 4. To my feeling, there should be mirror planes leading to a symmetry like 4mm rather than 4. If not, then what is breaking the mirror symmetries of the orthorhombic crystal? Is it possible that a ferroelectric multidomain state is responsible for the seemingly lower symmetry?

(5) How general are the results and the extraordinary multiferroic parameters obtained here? In other words, how can the concepts applied here be transferred to other compositions?

Smaller points:

(6) The style of Fig. 2a with data points and graphical elements on a background which is in itself a measured image of strongly fluctuating brightness is very confusing.

(7) The ferroelectricity of BiFeO₃ is described as displacive. It is not of the classical hybridization-driven BaTiO₃-like displacive soft-mode character, however, but caused by an electronic lone pair.

(8) Contrary to what the authors write in the abstract, there are other type-II room temperature multiferroics, like some hexaferrite compounds.

Reviewer #3 (Remarks to the Author):

The paper reported on the method of obtaining 3D straining in nanocomposite (SmMnO₃)_{0.5}((Bi,Sm)₂O₃)_{0.5} films. The basic idea is transparent and includes a high originality, and the paper is clearly written and well organized. The work shows how it is possible to achieve large in-plane compressive and out-of-plane tensile strains induced by the stiff (Bi,Sm)₂O₃ nanopillars embedded in the SmMnO₃. It is quite interesting idea. However, I am wondering the reason the authors claimed large room-temperature ferroelectricity and ferromagnetism with Curie temperature ~90 K? This work is very systematic and interesting as it presents a very good combination and blend of experimental results supported nicely by theoretical predictions for the induced strain levels to my view the paper is purposeful and can be recommended for acceptance. I have a feeling that the discussion section is a bit long and a large portion of it is really the X-ray RSM data that actually probably should belong to the results section. Furthermore, in the introduction what kind of applications in the sensors, that this materials with T_c (M) in a cryogenic temperature range can actually be used in, while the authors make a list of some other potential applications

Reviewers' comments:

Reviewer #1 (Remarks to the Author):

This manuscript describes the study of strain-induced property evolution in nanocomposites for SmMnO₃ with (Bi, Sm)₂O₃ nanopillars embedded in the SmMnO₃. As noted, materials like SmMnO₃ are known to be type-II multiferroics, but generally have ordering temperatures that are well below room temperature. Here, the production of large in-plane compressive and out-of-plane tensile strains are said to produce room-temperature polarization in the SmMnO₃. The magnetic order is found to be ferromagnetic with a critical temperature of ~90K while bulk SmMnO₃ is antiferromagnetic. Finally, density functional theory is used to explore the mechanism giving rise to the ferroelectric order which is stated to arise from a “modification of the exchange coupling.”

The main observations here are the large increase in the FE transition temperature (above RT vs. < 40 K in bulk), the large polarization (~4 uC/cm²), and the observation of FM order with a T_c ~ 90 K. At face value this seems quite dramatic. This said, there are a number of statements which could be better supported by experimental proof to confirm these claims. In particular, attention is drawn to measurements of ferroelectricity and the Mn-O-Mn bond angles. Both of these are considered lacking at this time and additional studies (described below) would be appropriate to improve these statements. As it stands, the paper has some speculative aspects at this time. Finally, the writing is disjointed, there are lots of small, short paragraphs. Overall it lacks the narrative form expected for a publication in a journal like Nature Communications. Thus, at this time, publication seems premature in a journal like Nature Communications.

Answer: The authors would like to thank the reviewer for the positive feedback and the critical comments on the submitted manuscript. The authors have considered every comments very carefully. After growing new films and doing more measurements, the authors have addressed them in the revised manuscript. The writing in the revised manuscript has been improved in line with the suggestions by the reviewer such as removing small, short paragraphs.

Detailed comments:

-Q1: Call the ferroelectricity “giant” is misleading and probably not appropriate. To be clear, polarization of 3.9 uC/cm² is not “giant” compared to regular ferroelectrics. This comes off a bit aggressive. It is noted that it is quite a bit larger than that observed in bulk SMO, but again it is not giant. This should be changed and removed from the paper.

Answer: The authors agree with the reviewer on this point. The authors were comparing the value to what has been measured in similar systems before.

Change in the manuscript: The authors accept the suggestion of the reviewer and removed the word ‘giant’ and have changed this to ‘large’.

- **Q2:** The section (on page 2) concerning the vertically aligned nanocomposites (VANs) is awkward for two reasons. First, it comes across as bit like a “sales pitch” (if you will) – strongly suggesting this is the only answer. Second, the bulleted list is odd and not something this Reviewer has seen in a large number of publications. This Reviewer would prefer a more passive, factual statement of approaches that have been used – advantages/disadvantages are fine – but less like a sales pitch for this approach.

Answer: Vertically aligned nanocomposites (VANs) are a unique way to create 3D strain. The in-plane strain originates by the substrate and out-of-plane strain originates due to the out-of-plane domain-matching epitaxy between the two phases. The authors agree with the reviewer that this may not be the only way (i.e. only answer) to create 3D strain but it is a very easy and controlled way. The relevant sentences have been revised accepting the reviewer’s suggestions.

The authors have revised the sentences to clearly state more passive and factual statements following the reviewer’s suggestion.

Change in the manuscript: Page 2, line 26-34. ‘proves’ changed to ‘shows’ in page 3, line 16.

- **Q3:** The statement “The film compositions are assumed to be the same as the target compositions used” seems out of place and unnecessary. First, without measurement such an assumption is unfounded. Second, it is becoming more clear that such assumptions are difficult to make – even in this system where the materials spontaneously form. Also, the paper itself notes that this specific system was picked because if some Bi ended up in the SMO it wouldn’t be a bad thing.

Answer: The authors would like to thank the reviewer for this critical comment on film composition. The authors agree with the reviewer that the sentence was unnecessary and has been removed in the revised manuscript.

Change in the manuscript: Page 3 (Growth of SMO:BSO VAN section)

- **Q4:** The language (page 3) “the film will be out-of-plane tensed” is not clear in meaning or consistent with literature discussions on such a matter.

Answer: The authors would like to thank the reviewer for the comment on out-of-plane tensile strain. Due to the domain matching epitaxy between SMO and BSO an out-of-plane strain on SMO by BSO is also generated vertically. The authors have revised the text for more clear understanding.

Change in manuscript: Page: 3, line 35, 41-42 (Section: Structural investigations of SMO:BSO VAN film by XRD and HRTEM).

- Q5: Why are the peaks so broad, rounded for the film in Fig. S1b?

Answer: Due to the orthorhombic structural nature of RMnO_3 , the in-plane XRD scan (i.e. phi scan) shows a broad phi scan peak as shown Fig. R1 (Ref. Thin Solid Films 516 (2008) 4899–4907)

Figure R1. YMnO_3 (Ref. Thin Solid Films 516 (2008) 4899–4907)

Additional broadening is caused by use of log scale on the Y-axis. The reason for which the Y-axis is plotted on a log scale is to accommodate the XRD peaks of the STO substrate in the same plot.

Hence, our data would be much less broad if plotted on a regular linear scale (Fig.R2).

Figure R2. Left is log-scale phi-scan and right is linear scale of one of our SMO:BSO films.

- Q6: Proof of ferroelectricity – The proof of ferroelectricity is some what questioned here. This is perhaps not a major fix, but the data provided seems an interesting choice. It is strongly recommended that the manuscript provide ferroelectric hysteresis loops for all samples – VANs and single films – and report the PUND data in a more traditional way – showing switching both as 1) a function of pulse time with different voltages and 2) as a function of voltage at different pulse times. The PUND data as shown is unprocessed and an odd choice. The Authors can find numerous examples of how such data is presented in the literature. The statement that SHG confirms ferroelectricity (page 3) is again misleading.

SHG probes noncentrosymmetry/polar structures – switching is not shown in the SHG data. So the SHG technically proves polarity, but not ferroelectricity. Finally, the authors are asked to refrain from using PFM as proof of ferroelectricity. Together with the other measurements it can support such a statement, but alone it is hard to say. There are many ways to get contrast like that imaged here from PFM.

Answer: The authors would like to thank the reviewer for the comments and suggestions on the presented ferroelectric data. The authors agree with the reviewer that the choice of presenting ferroelectric data is not very traditional. The SMO-BSO VAN films are somewhat leaky, similar to many other ferroelectric materials when in thin film form, particularly BFO-based (Hiroshi Naganuma, et. al., Appl. Phys. Express, 1, 061601 (2008); C. A-Paz de Araujo, et. al., Nature, 374, 627–629 (1995); M. Dawber, K. Rabe and J. Scott, Rev. Mod. Phys., 77, 1083, (2005)), which makes it extremely difficult to measure them in traditional way. This is the reason that many people show PFM images instead of P-E loops (Alexei Gruverman, et. al., Nature Communications, 10, 1661 (2019); Fukai Liu, et. al., Nature Communications, 7, 12357 (2016); Yu Tian, et. al. Nature Communications, 9, 3809 (2018); N. Balke, et. al., Nature Nanotechnology, 4, 868–875(2009)). Due to this reason, the authors have provided proof of room temperature ferroelectricity using multiple measurement methods. Detailed ferroelectric investigations using standard methods will be possible after work focused on improving film quality, which can take some time with the range of variables. Due to the same reason, the PUND data as shown is unprocessed. The main aim of the PUND is to show the switching of the polarisation and to estimate the polarisation value; and the present method serves that purpose.

In the revised Supplementary Information, the authors have provided ferroelectric hysteresis P-E loops for all three films (Fig. S3). Only the thick 100nm VAN film shows a ferroelectric hysteresis loop at room temperature. This matches well with the PFM and PUND data measured at room temperature, which show only the 100 nm film to be FE.

In addition, new PUND data with different pulse times (1 msec, 0.1 msec and 0.01 msec) have been included in the revised Supplementary Information (Fig. S4). For a 0.01 msec pulse time the amplitude of polarization is very low since the time is not enough to switch the ferroelectric polarization along the applied field direction. As the pulse time increases the polarization amplitude increases, as is expected in ferroelectric films.

In terms of voltage dependent PUND measurements, the authors note that increased voltages create high leakage in the films. The low voltage PUND gives partial switching. Hence, PUND with different voltages are not ideal and does not give any additional information in this case. [Ref: B: J.T. Evans, Characterizing Ferroelectric Materials, Radiant Technologies, Inc. March 7, 2011:

[https://www.ferrodevices.com/1/297/files/Ferroelectric_Properties_and_Instrumentation\(1\).pdf](https://www.ferrodevices.com/1/297/files/Ferroelectric_Properties_and_Instrumentation(1).pdf).

An alternative method to prove ferroelectricity is to do PUND measurements at the nanoscale (nano-PUND) in PFM. This is a relatively new method to directly probe the ferroelectric polarization at nanoscale level and specially for smaller polarization charge. By this method, it is possible to detect leaky and non-leaky area by PFM and do the nano-PUND measurement on non-leaky area of the sample. The new nano-PUND for 10V have been included in the revised manuscript in Supplementary information (Fig. S5a-c). By using same

technique, the authors also measured PE loops with two different frequencies (0.1 Hz and 0.5 Hz) i.e. two different pulse times and voltages (10 V and 5V), Fig. S5d. The low voltage (with voltages/field lower than coercive) PE loop did not produce any switching loop, as in the case of normal/standard Radiant method. The new data clearly proves ferroelectricity since the nano-PUND method is a direct measurement with reduced parasitic capacitance effects due to the direct measurement i.e. less circuits.

The authors agree with the reviewer on SHG. The electric dipolar SHG probes non-centrosymmetry in materials. Any property that breaks the inversion symmetry in the materials could contribute to SHG including polar lattice and spontaneous polarization. In this study, the authors combine SHG, PFM with PUND to show that there is large ferroelectricity at room temperature. The authors have now also undertaken measurements on the temperature dependence of the SHG to show a non-centrosymmetric to centrosymmetric transition (Fig. S6).

The authors also agree with the reviewer that PFM measurements alone cannot be the proof of ferroelectricity. The combination of PFM phase contrast, PE hysteresis loops with different voltages, PUND data with different pulse times and SHG collectively prove the room temperature ferroelectricity. The authors have reframed the sentences as per the reviewer's suggestion.

Change in manuscript: Page: 6 (Yellow highlighted lines)

New ferroelectric hysteresis loops, time dependent PUND data, nano-PUND data and voltage/time dependent PE loops are added in the Supplementary Information (Fig. S4 & 5).

- Q7: Ferroelectric measurements – Please provide details on how the measurements were done.

Answer: The authors would like to thank the reviewer for the suggestions. The methods section has been revised following reviewer's suggestions.

Changes in the manuscript: Page 14, line 13-14, 30-34.

For the hysteresis loops, please provide frequency dependent studies and show that they follow expected trends.

Answer: The authors have provided new frequency dependent measurements by nano-PUND in the Supplementary Information, Figure S5d. The frequency clearly does not affect the polarization value as long as it is above the switching voltage. A noticeable decrease in coercivity with decrease of frequency is observed.

Changes in the manuscript: Figure S5d in the Supplementary Information.

The biggest question here is how the area was determined considering that the material is not homogenous either in-plane or out-of-plane, thus some error bars on the value of polarization seem essential.

Answer: The authors agree with the reviewer that it is difficult to determine the exact area of the ferroelectric material, SMO, in the VAN structure. To simplify this, the authors consider the area of electrode, which also includes non-ferroelectric BSO. Since the matrix phase (in this case BSO) is an intrinsic part of the VAN structure, the authors consider this is the best possible way to represent the polarisation per unit area, and it represents a *minimum* polarisation value. A more clearly defined VAN structure (using a different pillar material to BSO, which does not over-grow the matrix material) would allow the authors to determine the area more precisely. This is a topic of future work.

The authors have included relevant discussion on this in the revised manuscript.

Changes in the manuscript: Page 14, line 25-29.

-Q8: SHG – It is stated that SHG was done to extract the point group symmetry of the VANs, but that point group symmetry is not given. A point group of 4 is provided – is this the full symmetry? No other symmetry elements? This symmetry needs to be self-consistent with the proposed structural changes later, is it? Second, SHG provides an excellent way to extract the ferroelectric transition temperature – this not done, but should be to further confirm the attribution of ferroelectric order. Switching and a transition temperature will go a long way to show this.

Answer: The authors would like to thank the reviewer for this excellent comment. The authors have revisited the SHG modelling, and the 4mm structure fits the data well. Any other point groups which have lower symmetry than point group of 4mm would also fit the SHG results. New temperature dependent SHG is done following the reviewer's suggestion, which confirms the transition from non-centrosymmetric to centrosymmetric structure at ~360K-370K.

Changes in the manuscript: Page 6, line 33-39.

Revised SHG model in Supplementary Information (yellow highlighted).

- Q9: Are the authors sure that the elongation seen in Figure 4 is splitting of a peak and not either measurement artifact, partial relaxation, or some combination thereof? Where should the strained and relaxed peak positions be for the o-SMO in these scans? Can this be marked?

Answer:

The authors understand the Reviewer's concerns about the elongation observed in XRD. In this study, a Panalytical Empyrean high-resolution XRD system was used. In this system, it is possible to choose several modes for RSM, quick or slow scan, or high or low accuracy depending on the demands of experiment. For a high intensity mode using a position sensitive detector (frame grab mode) an instrument artifact elongation appears around the substrate peak along the diagonal direction. Figure R3 (a) shows an RSM measured in this mode on a plain LSMO/STO film.

Figure R3. RSM of LSMO film on (001) STO (a) RSM around (103) STO captured by the frame grab mode and (b) RSM around (113) STO captured by the Parallel beam optics with a proportional counter.

On the other hand, Figure R3 (b) shows an RSM of the same LSMO/STO film with parallel beam optics (proportional counter). Here, no measurement artifact is observed like Figure R3 (a). A slight elongation can be observed around the substrate. Therefore, the elongation of RSM data of Figure 4 is the tail of film peak not from an instrumental artifact. (NB: In addition, to obtain more accurate lattice parameters, negative offset mode was used).

Regarding the strained and relaxed peak positions, it is possible to identify these in the case of the thick (100 nm) VAN film as peak separation along the in-plane direction (i.e. along Q_x in the RSM) is clear. In the RSM, the peak close to STO corresponds to a more strained film region than the peak relatively far from STO because ‘strained’ means ‘strained by STO’ and ‘similar in-plane lattice parameter of pseudo-cubic SMO as STO’.

To answer the question about where the peak positions should be for strained and relaxed phases:

First, let take the 20 nm VAN film. Here, there is very little relaxation as the film is thin. For the 100 nm VAN film, the film does not relax as a normal plain film would because of the influence of the nanopillars on controlling the strain. Hence, instead of strain relaxation with increasing film thickness, a new phase appears, i.e. *o*-SMO2. This VAN strain, which produces *o*-SMO2, is effective when there less of a substrate effect (i.e. when the film is thicker).

For the 100 nm plain SMO film, there *will be* strain relaxation. The main relaxation is to the R phase (as discussed already in the Supplementary Information and shown in Fig. S1a). This relaxed R phase peak is not within the RSM window. However, a bulging downwards of the SMO peak below the STO peak is also observed for the 100 nm. This bulging likely occurs as the out-of-plane lattice parameter of SMO tends towards the d (110) bulk value which is larger than the out-of-plane value for SMO when strained on STO. Hence the out-of-plane lattice parameter increases from 7.805 Å (thin (20 nm) VAN film strained by STO), towards 7.8929 Å (d (110) bulk).

Since, the strain relaxation has already been discussed in the Supplementary Information, the bulging effect is minor, and not relevant for the VAN films where the interesting effects occur, the authors do not add extra information about this in the revised manuscript. It will likely detract from the main story.

- Q10: The schematics in Figure 5 (and the associated text) are a bit hard to follow. There is a big leap of faith to be made to believe this is the “full” picture. Can direct measurements in the STEM be done to prove this? Furthermore, if the structure/strain is varying throughout, can we be sure about the extracted/reported values of magnetism? Polarization? Aren't they likely to also be changing?

Answer: The authors thank the reviewer for these comments and agree that further clarification is needed to make the points more clear. Figure 5 has been drawn based on the XRD data presented in the Table S1. This is also supported by STEM image of Figure S2. The change strain level in SMO-2 from SMO-1 is 2-3%. There is a phase transition from SMO-1 to SMO-2, which occurs at around a critical thickness ~15 nm, as determined from the magnetic measurement and STEM images. The different phases SMO1 and SMO2 are clearly observed in the calculated strain maps in Fig. S2 bi and bii). The in-plane strain builds gradually as the film thickens. This has been now clearly shown in the revised schematic diagram.

Change in the manuscript: The schematic figure (Figure 5) has been modified more accurately based on the results for better understanding and to make it easy to follow.

The magnetisation value changes from 30 emu/cc in SMO1 to 100 emu/cc in SMO2. This has been shown in the inset figure of Figure 3 and discussed under the Magnetic Properties of VAN films section in page 11 (line 33-43). The polarization also change from non-ferroelectric in SMO-1 to ferroelectric in SMO-2. The text has been revised to make this more clear.

Change in the manuscript: Page: 11, line 14-15.

- Q11: The assertion that the Mn-O-Mn bond angle is not based on a measurement, but is deduced from the structural changes. As such it is a bit speculative. Direct proof of this would be nice.

Answer: The authors agree with the reviewer that the Mn-O-Mn bond angle is deduced by using DFT calculation. For this calculation, the experimental data of lattice constants measured by XRD were used. This is one of the most convenient and efficient way to

estimate the value of the Mn-O-Mn bond angle [Ref: T. Aoyama, et. al., Nature Communications, volume 5, Article number: 4927 (2014)]. The authors agree with the reviewer that experimental proof of the Mn-O-Mn bond angle would be nice. Unfortunately, this is not possible experimentally because of the complex 3D structure as well as gradual change in strain, and hence the DFT calculations are the best way possible at present for the authors to estimate the bond angles in this complex 3D structure.

Reviewer #2 (Remarks to the Author):

Obtaining large and high-temperature magnetization and polarization in multiferroics is still a major challenge. Working with composite materials may help to avoid this dilemma, but because of their grainy, irregular structure any magnetoelectric coupling in these is usually mediated by periodic deformation and therefore an AC effect occurring under specific circumstances only. With their work, the authors bring together the advantages of crystalline and composite multiferroics. They grow a SmMnO₃/(Bi, Sm)2O₃ nanopillar composite, yet as epitaxial film with well-defined interfaces between the constituents. They thus obtain above-room temperature ferroelectricity with unusually high spontaneous polarization and a sizable magnetization up to almost 100 K. Key to these pronounced multiferroic properties are the high strain values they can stabilize by the nanopillar architecture.

This work demonstrates once more that the toolbox for obtaining pronounced multiferroic responses towards higher temperature is not yet empty. I find the approach of using epitaxial nanocomposites original and useful and the performance parameters obtained with it remarkable. To my opinion, this work deserves publication in Nature Communications.

There are a couple of open issues the authors should work on prior to acceptance.

Answer:

The authors would like to thank the reviewer for the positive feedback on the manuscript. The authors have addressed all the points raised by the reviewer and done required modifications in the revised manuscript.

Q (1) My main point is that I do not quite understand the relation between ferromagnetic order and the cluster-glass-like behavior. From the text, I am not sure in what temperature range exactly is the cluster-glass-like response is observed. So how does it interfere with the notion of ferromagnetism: Can one talk about ferromagnetism at all if a cluster-glass state is present?

Answer: The authors would like to thank the reviewer for this critical comment. Cluster glass-like behaviour is observed in these perovskite systems, when a frustration between the antiferromagnetic (AFM) and ferromagnetic (FM) ordering occurs. These interactions arise when the Mn–O–Mn bond angle and bond length gives rise to intermediate interactions between antiferromagnetic 180° super exchange and ferromagnetic 90° super exchange [Ref.

J. M. D. Coey, et. al., Adv. Phys. 48(2) (1999), pp. 167–293; J. Wu, et. al., Phys. Rev. Lett. 94 (2005), pp. 037201–037204; S. Mukherjee, et. al. Phys. Rev. B 54, 9267 (1996)]. Due to cluster-like behaviour, the ZFC magnetic moment or susceptibility reach maximum values ($T_{\max} \sim 20\text{K}$). This also indicates the coexistence of both near-neighbour AFM and FM interactions. Due to the induced strain, the decreased Mn-O in-plane bond length means that FM interactions are favoured which increases the ferromagnetic curie temperature ($T_{C,FM}$). The unsaturated magnetic hysteresis loop also signifies the coexistence of FM and AFM interactions at the low temperature. [Ref: J. A. Alonso, J. L. Martínez, M. J. Martínez-Lope, M. T. Casais, and M. T. Fernández-Díaz; Phys. Rev. Lett. (1999) 82, 189] The authors have modified the sentences for better understanding.

Change in the manuscript: Page 7, line 11-14.

Q (2) What is the ferroelectric Curie temperature? As the spontaneous polarization is detected by second harmonic generation (SHG), above-room-temperature measurements up to $T(C)$ should be possible because thermal conductance is not detrimental to SHG.

Answer: The authors would like to thank the reviewer for the comment on the ferroelectric Curie temperature. The authors have done new temperature dependent SHG measurements. The data indicates that the non-centrosymmetric to centrosymmetric transition is broad, with a possible onset of long-range order setting in at $\sim 360\text{K}$ - 370K . The new information has been included in the revised manuscript.

Changes in the manuscript: Page 6, line 34-39.

Temperature dependent SHG measurement is added in the revised Supplementary Information (Fig. S6).

Q (3) Is it possible that the polarization is caused by interface charges rather than by the onset of ferroelectricity? Interface charges may be switchable, too, and are not necessarily detected by a PUND measurement.

Answer: The authors agree with the reviewer that the polarization can be caused by the interface charge. But the interface charge is not stable. The PFM measurements were done overnight multiple times and no changes in the polarisation were found. Similar measurements were also done on thin VAN film and SMO films to crosscheck. No switchable polarisation (write and rewrite was observed) for these films. Also, interfacial charge should not produce switchable remanence (P_r) and should not decrease for fast switching (i.e. 0.01 ms pulse in Supplementary Information) in PUND measurements. Hence, these measurements prove that the polarisation is not caused by interface charges. Further the new nano-PUND measurement confirm the observation.

Q (4) The point symmetry of the film is concluded from the SHG measurements as 4. To my feeling, there should be mirror planes leading to a symmetry like 4mm rather than 4. If not, then what is breaking the mirror symmetries of the orthorhombic crystal? Is it possible that a ferroelectric multidomain state is responsible for the seemingly lower symmetry?

Answer: The authors appreciate this excellent comment, which the earlier referee also raised. The authors have revisited the SHG modelling, and 4mm fits equally well. Any other point groups which have lower symmetry than the point group of 4mm would also fit the SHG results. The new temperature dependent SHG measurements confirm the transition from non-centrosymmetric to centrosymmetric structure at $\sim 360\text{K}$ - 370K .

Changes in the manuscript: Page 6, line 33-39. Figure 2d has been revised.

Revised SHG model in Supplementary Information (yellow highlighted).

Q (5) How general are the results and the extraordinary multiferroic parameters obtained here? In other words, how can the concepts applied here be transferred to other compositions?

Answer: The concept is certainly applicable to other materials systems. The authors have studied several other material systems such as EuTiO_3 - Eu_2O_3 where sizeable multiferroic properties have been changed due to 3D strain (paper in press).

Smaller points:

Q (6) The style of Fig. 2a with data points and graphical elements on a background which is in itself a measured image of strongly fluctuating brightness is very confusing.

Answer: The authors apologise for this. The figure has now been revised according to the reviewer's suggestion and this has improved the clarity of Fig. 2a greatly.

Q (7) The ferroelectricity of BiFeO_3 is described as displacive. It is not of the classical hybridization-driven BaTiO_3 -like displacive soft-mode character, however, but caused by an electronic lone pair.

Answer: The authors thank the reviewer for pointing out this unforced mistake. The text has been revised according to the reviewer's suggestion.

Change in the manuscript: Page: 2, Line 1

Q (8) Contrary to what the authors write in the abstract, there are other type-II room temperature multiferroics, like some hexaferrite compounds.

Answer: The authors agree with the reviewer that there are other type-II room temperature multiferroics such as hexaferrites. However, there are no reported type-II room temperature multiferroics of orthorhombic structure. The text has been revised to clear this confusion.

Change in manuscript: Page 1, line19.

Reviewer #3 (Remarks to the Author):

The paper reported on the method of obtaining 3D straining in nanocomposite $(\text{SmMnO}_3)_{0.5}(\text{Bi,Sm})_2\text{O}_3)_{0.5}$ films. The basic idea is transparent and includes a high originality, and the paper is clearly written and well organized. The work shows how it is possible to achieve large in-plane compressive and out-of-plane tensile strains induced by the stiff $(\text{Bi,Sm})_2\text{O}_3$ nanopillars embedded in the SmMnO_3 . It is quite interesting idea. However, I am wondering the reason the authors claimed large room-temperature ferroelectricity and ferromagnetism with Curie temperature ~ 90 K? This work is very systematic and interesting as it presents a very good combination and blend of experimental results supported nicely by theoretical predictions for the induced strain levels to my view the paper is purposeful and can be recommended for acceptance. I have a feeling that the discussion section is a bit long and a large portion of it is really the X-ray RSM data that actually probably should belong to the results section. Furthermore, in the introduction what kind of applications in the sensors, that this materials with T_c (M) in a cryogenic temperature range can actually be used in, while the authors make a list of some other potential applications

Answer: The authors would like to thank the reviewer for the positive feedback and the excellent comments. The authors have considered every comments very carefully and addressed them in the revised manuscript.

Comment: However, I am wondering the reason the authors claimed large room-temperature ferroelectricity and ferromagnetism with Curie temperature ~ 90 K?

Answer: The authors would like to thank the reviewer for raising this question. Bulk SmMnO_3 is para-electric at room temperature and the ferromagnetic $T_{C,FM}$ is < 40 K. The amount of polarization $2P_R \approx 3.9 \mu\text{C}/\text{cm}^2$ of SMO VAN is higher than its bulk value. The new SHG measurement confirm the $T_{C,FE}$ is ~ 360 K- 370 K. This room temperature, strong ferroelectric system with ferromagnetic $T_{C,FM}$ at ~ 90 K has been observed in our VAN SMO-BSO system due to the 3D-strain induced by BSO on SMO.

Comment: I have a feeling that the discussion section is a bit long and a large portion of it is really the X-ray RSM data that actually probably should belong to the results section.

Answer: The authors agree with the reviewer that a significant part of the discussion is on X-ray RSM data. However, since the observed ferroelectric and ferromagnetic behaviour arises from structural changes, resulting from the strain, the authors think it is important to include this information in the discussion. If the reviewer strongly feels the authors need to put more information in Supplementary Information, that can be certainly done.

Comment: Furthermore, in the introduction what kind of applications in the sensors, that this materials with T_c (M) in a cryogenic temperature range can actually be used in, while the authors make a list of some other potential applications.

Answer: The authors would like to thank the reviewer for the excellent suggestions. The authors have included a relevant discussion in the revised manuscript considering the reviewer's suggestion. The authors agree that magnetic sensors operating at liquid N_2 are not

ideal, although there are reports of such devices, e.g. Keiji Tsukada, et. al., AIP Advances 7, 056670 (2017). However, the authors have removed the mention of sensors so as to not cause confusion here. Instead, the authors have added potential applications in memory devices as suggested by the reviewer.

Change in manuscript: Page 2, Line 2-3

Reviewers' comments:

Reviewer #1 (Remarks to the Author):

This reviewer has carefully read the responses and the revised manuscript. Overall, the authors are commended for a thoughtful response to the (numerous) comments. This is clearly complex work and thus solicited numerous questions from this reviewer. In most cases, the authors' responses and edits are thought to be adequate to address the questions. In turn, the manuscript has been improved. This said, concerns about the ferroelectricity remain.

First, the reviewer must disagree with the rebuttal letter – the data in Figure S3 does not show evidence of ferroelectricity. Loops like this are, in turn, the reason that the attribution and reporting of a polarization value from the PUND seem unjustified. The reviewer appreciates the sincerity with which the authors have taken this comment and attempted to measure the effect. This said, the new PUND data at different pulse times also draws concern. First, there is nothing intrinsic to switched polarization that suggests this 0.01 msec time scale should not switch. Second, generally one would have to increase the voltage (perhaps) to get the polarization to switch at the faster pulse times, but the voltage was already very large to begin with – thus begging the question of why a similar value was not measured at the faster pulse times? The varying PUND data leads this reviewer to ask if the measured effects at slower speeds can truly be (100%) attributed to ferroelectric polarization (as opposed to some polarization mixed with charge carrier motion in the leaky films)? If this is the case (or even potentially so), then reporting the value of "large" polarization is unjustified at this time. While the reviewer is fully sympathetic to the challenges of this study, the reviewer is also sensitive to assuring accurate information about such matters is presented in the literature. As it stands, it is the opinion of the reviewer that it is likely that the authors have sufficiently proven "ferroelectric switching" but providing a quantitative measure of the polarization seems unfounded at this time. It is strongly suggested that the "large" polarization values be downplayed or removed – the data are rather questionable in this regard. The reviewer – in light of the rest of the data – supports the statement that ferroelectric order is present. This is further convoluted by the lack of knowledge (at no fault of the authors) of the area and with the varying polarization with strain variations in the structures – so the quantitative value seems somewhat meaningless. This is a call for the authors to make, but caution is recommended.

Reviewer #2 (Remarks to the Author):

The authors answered very profoundly and convincingly to my comments and, to my impression, also to those in the (rather intriguing) report of Reviewer 1. I therefore recommend publication.

Reviewer #3 (Remarks to the Author):

I think that the authors responded and addressed adequately most of the questions. I believe that reporting results highlighting the path for obtaining high ferroic properties via strain-induced coupling in nanocomposites SmMnO₃ with (Bi, Sm)₂O₃ nanopillars embedded in the SmMnO₃ are very interesting. It was very instructive to see the measurement of the standard P-E loop to recognize "real" ferroelectricity as claimed. I think that result were presented(in the Supplementary section) even though lossy character of the materials PE response might still be a problem especially in a light of any further applications. Fundamentally, the way of getting a gain in in-plane strain using epitaxial nanocomposites is very novel and I am positive this work deserves publication in Nature Communications.

Reviewers' comments:

Reviewer #1 (Remarks to the Author):

This reviewer has carefully read the responses and the revised manuscript. Overall, the authors are commended for a thoughtful response to the (numerous) comments. This is clearly complex work and thus solicited numerous questions from this reviewer. In most cases, the authors' responses and edits are thought to be adequate to address the questions. In turn, the manuscript has been improved. This said, concerns about the ferroelectricity remain.

Comment: First, the reviewer must disagree with the rebuttal letter – the data in Figure S3 does not show evidence of ferroelectricity. Loops like this are, in turn, the reason that the attribution and reporting of a polarization value from the PUND seem unjustified. The reviewer appreciates the sincerity with which the authors have taken this comment and attempted to measure the effect. This said, the new PUND data at different pulse times also draws concern. First, there is nothing intrinsic to switched polarization that suggests this 0.01 msec time scale should not switch. Second, generally one would have to increase the voltage (perhaps) to get the polarization to switch at the faster pulse times, but the voltage was already very large to begin with – thus begging the question of why a similar value was not measured at the faster pulse times? The varying PUND data leads this reviewer to ask if the measured effects at slower speeds can truly be (100%) attributed to ferroelectric polarization (as opposed to some polarization mixed with charge carrier motion in the leaky films)? If this is the case (or even potentially so), then reporting the value of “large” polarization is unjustified at this time. While the reviewer is fully sympathetic to the challenges of this study, the reviewer is also sensitive to assuring accurate information about such matters is presented in the literature. As it stands, it is the opinion of the reviewer that it is likely that the authors have sufficiently proven “ferroelectric switching” but providing a quantitative measure of the polarization seems unfounded at this time. It is strongly suggested that the “large” polarization values be downplayed or removed – the data are rather questionable in this regard. The reviewer – in light of the rest of the data – supports the statement that ferroelectric order is present. This is further convoluted by the lack of knowledge (at no fault of the authors) of the area and with the varying polarization with strain variations in the structures – so the quantitative value seems somewhat meaningless. This is a call for the authors to make, but caution is recommended.

Reply: The authors would like to thank the reviewer for positive comments on the revised manuscript and valuable suggestions on the quantitative value of polarization, which the authors fully agree with. The authors also agree with the reviewer that the data in Figure S3 does not show evidence of ferroelectricity on its own. Hence, the authors have provided multiple evidences of room temperature ferroelectricity.

The authors understand the reviewer's concern on low polarisation value at the faster pulse time (0.01 ms) in PUND measurements. The slight decrease of polarization from 1 ms to 0.1 ms pulse time is common in ferroelectric thin films. [Nature Communications 10, 1282 (2019); Appl. Phys. Lett. 86, 092905 (2005); RSC Adv. 6, 30148-30153 (2016); Materials, 10, 1318 (2017)] The authors agree with the reviewer that there is no intrinsic reason for the film to not switch polarization for 0.01 ms time scale at 9V. The pulse time of 0.01 ms is

below the minimum pulse width (50 μs or 0.05 ms) of the instrument. Hence, the polarization value drops significantly for 0.01 ms pulse width. This has now clearly stated in the revised manuscript. The authors would like to thank the reviewer for the comment, which has helped the authors to explain the reason of such a low polarization value for the 0.01 ms timescale.

Change in manuscript: Supplementary Information, Page 4, line 20-22. Yellow highlighted.

The authors fully agree with the concern of the reviewer in determining the value of polarization. It is difficult to determine such a value of polarization precisely in this complex 3D vertically aligned nanostructure due to several reasons. This was also explained in the previous reply and revised manuscript. This is now further explained in the revised manuscript with more clarity.

Change in manuscript: Supplementary Information, Page 4, line 22-23. Yellow highlighted.

The authors have removed the word 'large' and downplayed the value of ferroelectric polarization in the revised manuscript considering the reviewer's suggestion. The authors would like to thank the reviewer for this suggestion.

Changes in manuscript:

Removed 'Large' from the title, line 9 & 14 in page 3, line 2 of page 13 (conclusion)

Abstract: Page 1, Lines 23-24 are revised.

Revised line 5 of page 3 (Deleted: A very high FE polarization ($P \approx 3.9 \mu\text{C}/\text{cm}^2$) is also achieved.) and line 15-16 of page 6

REVIEWERS' COMMENTS:

Reviewer #1 (Remarks to the Author):

The Reviewer agrees with the edits and recommends publication.